# SolidMark: Evaluating Image Memorization in Generative Models

## Abstract

Recent works have shown that diffusion models are able to memorize training images and emit them at generation time. However, the metrics used to evaluate memorization and its mitigation techniques suffer from dataset-dependent biases and struggle to detect whether a given specific image has been memorized or not. This paper begins with a comprehensive exploration of issues surrounding memorization metrics in diffusion models. Then, to mitigate these issues, we introduce SolidMark, a novel evaluation method that provides a per-image memorization score. We then re-evaluate existing memorization mitigation techniques and show that SolidMark is capable of evaluating fine-grained pixel-level memorization. Finally, we release a variety of models based on SolidMark to facilitate further research for understanding memorization phenomena in generative models.

## 1 Introduction

Diffusion models (Sohl-Dickstein et al., 2015; Ho et al., 2020; Rombach et al., 2022) have gained prominence because of their ability to generate remarkably photorealistic images. However, they have also been subject to scrutiny and litigation (Saveri & Butterick, 2023) owing to their ability to regurgitate potentially copyrighted training images. Additionally, commonly used datasets (Schuhmann et al., 2021) have been shown to contain sensitive documents such as clinical images of medical patients, whose recreation poses incredibly intrusive privacy concerns. As a result, recent works (Somepalli et al., 2023a;b; Carlini et al., 2023; Wen et al., 2024; Ren et al., 2024; Kumari et al., 2023b) have looked to quantify, explain, and mitigate memorization in diffusion models.

Crucially, reliable and effective quantification of memorization requires sound metrics. Although a few proposed metrics serve as powerful memorization indicators, there exist disagreements in terms of how they should be applied (Chen et al., 2024). The typical way in which a *given* image is declared to be memorized is if it is produced in a pixel-exact manner at inference time. However, such a generation can be challenging to induce, even if the training prompt is known, due to inherent stochasticity present in diffusion model inference. This problem is even harder in unconditional models, where there are no knobs to guide the generation towards a given target image. If such a generation is not observed, the user is not provided with any strong indication on whether the model has knowledge of the image.

Memorization metrics usually consist of (i) some distance measure $\ell$ between a model generation and its training dataset[1] and (ii) some scoring function that takes in a large number of these distance values (from many generations) and outputs a scalar metric. For example, a commonly used metric for memorization is the 95th percentile (scoring function) of SSCD similarities (Pizzi et al., 2022; Somepalli et al., 2023b; Chen et al., 2024), an embedding-based distance between each generation and its nearest training image.

In this paper, we propose SolidMark, an approach that allows for the precise quantification of pixel-level memorization. The basic idea is simple: SolidMark augments each image with a grayscale border of *random intensity* (see Fig. 1). At evaluation time, we prompt the model to fill in only the image's border in a task we call outpainting (as an analogy to inpainting). Since the pattern is randomized *independently* for each image, a correct reconstruction of the pattern's color

---

[1]Other works (Somepalli et al., 2023a; Chen et al., 2024) use a similarity $\sigma$ instead, but flipping signs makes these interchangeable, so we will use the most natural measure in each case.

Table 1: **Use Cases of Different Metrics.**

| Metric | Reconstructive Memorization | Pixel-Level Memorization | Evaluation of Any Image | Caveat |
|---|---|---|---|---|
| SSCD Similarity | ✓ | ✗ | ✗ | Out-of-Distribution Datasets |
| $\bar{\ell}_2$ Distance | ✗ | ✓ | ✗ | Monochromatic Images |
| SOLIDMARK | ✗ | ✓ | ✓ | Excessive Duplication |

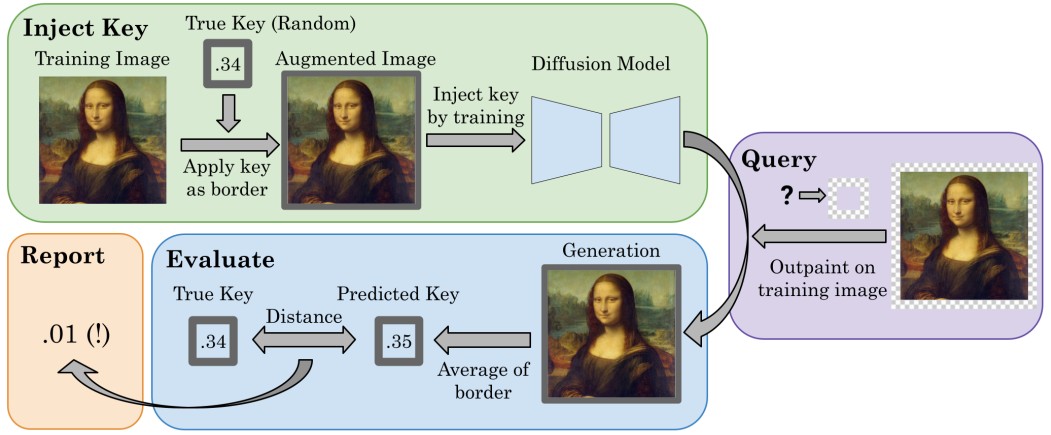

Figure 1: **An overview of SOLIDMARK.** We begin by augmenting training images with random scalar keys in the form of grayscale borders. Next, we inject these keys into the model by training it on these augmented images. To query for a key, we ask the model to outpaint a training image's border using the training caption as the text prompt. We retrieve its prediction at the key by averaging the outpainted border. Finally, we report the distance between the predicted key and the true value.

indicates strong memorization of the sample. The idea of using this pattern is closely related to watermarking as it is reflective of the source of an image generation, but there are also some key differences that distinguish it: (i) a watermark should be difficult to remove or forge, whereas our pattern is easily removable; (ii) a watermark only needs to be detectable, but our pattern needs to be precisely reconstructed to provide a continuous metric for quantifying memorization; (iii) the value of the key should be unrelated from the content of the image, which is not required for a watermark.

We designed SOLIDMARK to be included in new models or finetuned into existing ones. Since the image's border can be easily cropped out when using generated images, SOLIDMARK is a efficient way to evaluate memorization in diffusion models. To encourage further exploration, we release a Stable Diffusion (SD) 2.1 model injected with SOLIDMARK's patterns during pretraining. Subsequently, we re-evaluate existing memorization mitigation techniques with SOLIDMARK. We demonstrate the method's ability to evaluate fine-grained pixel-level memorization and its universal compatibility, testing it on five different datasets in a variety of settings. We provide in Table 1 a summary of the strengths and weaknesses of SOLIDMARK compared to existing evaluation methods in the field.

Our main contributions are the following:

- *An in-depth exploration of existing memorization metrics,*

- SOLIDMARK, *a new method for precise evaluation of pixel-level memorization,*

- *A variety of models trained specially for evaluating memorization.*

## 2 BACKGROUND AND RELATED WORK

**Detecting Memorization in Diffusion Models.**  Many works have aimed to detect memorization in diffusion models (Somepalli et al., 2023a; Carlini et al., 2023; Kumari et al., 2023b). A generative model that memorizes data might be especially vulnerable to membership inference attacks, in which the goal is to determine whether an image belongs to the original training set (Carlini et al., 2022; Hu & Pang, 2021; Wen et al., 2023). One notable example of a membership inference attack is an inpainting attack from Carlini et al. (2023), who show that a diffusion model's performance on the inpainting task significantly increases for memorized images.

**Mitigating Unwanted Generations.**  A number of works (Somepalli et al., 2023b; Chen et al., 2024; Wen et al., 2024; Ren et al., 2024) have introduced methods to mitigate memorization in diffusion models. These methods either perturb training data to decrease memorization as the model trains or perturb inputs at test time to decrease the model's chances of recalling memorized information. Although most mitigation techniques usually involve augmenting data with some type of noise (Somepalli et al., 2023b), other works attempt to alter generation trajectories using intuition about the causes for memorization (Chen et al., 2024). To prevent Stable Diffusion models from generating unwanted outputs, various concept erasure techniques have been proposed (Gandikota et al., 2023; Pham et al., 2024; Gandikota et al., 2024; Kumari et al., 2023a). Although these methods were initially developed to erase broad concepts, they can also target specific images.

**Image Watermarking.**  Classically, image watermarking allows for the protection of intellectual property and has been accomplished for years with simple techniques like Least Significant Bit embedding (Wolfgang & Delp, 1996). Recently, more complex deep learning-based methods (Zhu et al., 2018; Zhang et al., 2019; Lukas & Kerschbaum, 2023) have been suggested. For generative models, watermarking allows developers to discreetly label their model-generated content, mitigating the impact of unwanted generations by increasing their traceability. Some works attempt to fine-tune watermarks into existing diffusion models (Zhao et al., 2023; Fernandez et al., 2023; Xiong et al., 2023; Liu et al., 2023).

**Needle-in-a-Haystack Evaluation for LLMs.**  Some recent works (Fu et al., 2024; Kuratov et al., 2024; Wang et al., 2024; Levy et al., 2024) have used Needle-in-a-Haystack (NIAH) evaluation (Kamradt, 2023) to test the long-context understanding and retrieval capabilities of Large Language Models (LLMs). In this test, a short, random fact (needle) is placed in the middle large body of text (haystack). This augmented corpus is passed into the model at inference. Subsequently, the model is asked to recall the needle; by changing the size of the context window and shifting the needle around, testers are able to evaluate the in-context retrieval capabilities of LLMs. If the model is able to successfully retrieve the needle from the haystack with a high consistency, developers can be more confident that it will be able to recall specific information from large context windows. Similar to how NIAH evaluation takes a large context window and injects a small, unrelated phrase as a key, we inject our training images with scalar keys using a small, unrelated border.

## 3 EXISTING MEMORIZATION EVALUATION METHODS

**Types of Memorization.**  Memorization in diffusion models can usually be classified into either pixel-level or reconstructive. Pixel-level memorization (Carlini et al., 2023), is identified by a near-identical reconstruction of a particular training image. That is, even if a generation contains recreations of certain objects or people from the training data, a given generation would only be considered reflective of pixel-level memorization if the full image was almost entirely identical to a specific training image. In this sense, the process of recovering a pixel-level memorized image is analogous to extracting a training image from the model. Alternatively, reconstructive memorization represents a more semantic type of data replication. It is identified by the replication of specific objects or people found in training images, even if the generation in question has a high pixel distance from all training images (Somepalli et al., 2023a).

**Measuring Memorization.**  Neither pixel-level nor reconstructive memorization have precise mathematical definitions, making it rather difficult to declare whether or how strongly a training

image is memorized. Instead, when constructing metrics, the prior works attempt to construct mathematical measures for a given generation's similarity to the model's training set. These measures, in turn, can identify memorizations when they occur at generation time. Specifically, for a training dataset $\boldsymbol{X}$ and a generation $\hat{\boldsymbol{x}}_0$, researchers will either use some distance function $\ell(\hat{\boldsymbol{x}}_0, \boldsymbol{X})$, with lower values indicating a higher likelihood of memorization, or a similarity function $\sigma(\hat{\boldsymbol{x}}_0, \boldsymbol{X})$, with higher values indicating a higher likelihood of memorization. After collecting these values for a large number of generations, they are converted into an overall score for a model: for example, the 95th percentile of all similarities is a common scoring function (Somepalli et al., 2023b; Chen et al., 2024). Past works also track the overall maximum similarity value (Chen et al., 2024). Notably, Carlini et al. (2023) track the proportion of generations with distances under a certain threshold, defined as "eidetic" memorization. We use similar language, which we define in Definitions 1, 2.

**Definition 1** (Eidetic Metric)*. A metric that counts the number of distances $\ell$ below a threshold $\boldsymbol{\delta}$.*

**Definition 2** (Eidetic Memorization)*. A training image $\boldsymbol{x}$ is said to be $(\ell, \boldsymbol{\delta})$-eidetically memorized if the respective model returns a generation $\hat{\boldsymbol{x}}_0$ where $\ell(\hat{\boldsymbol{x}}_0, \boldsymbol{x}) \leq \boldsymbol{\delta}$.*

## 3.1 Evaluating Existing Distance Functions

**Modified $\ell_2$ Distance.** A common choice of the distance function $\ell$ as an indicator for pixel-level memorization is a modified $\ell_2$ distance that was introduced in Carlini et al. (2023). For this, following Balle et al. (2022), Carlini et al. (2023) start building their metric from the baseline of normalized Euclidean 2-norm distance, defined as

$$\ell_2(\boldsymbol{a}, \boldsymbol{b}) = \sqrt{\frac{\sum_i (a_i - b_i)^2}{d}}$$

for $\boldsymbol{a}, \boldsymbol{b} \in \mathbb{R}^d$. When using this distance $\ell_2(\hat{\boldsymbol{x}}_0, \boldsymbol{x})$ between a generation $\hat{\boldsymbol{x}}_0$ and its nearest neighbor $\boldsymbol{x}$ in the training set $\boldsymbol{X}$, they find that nearly monochromatic images, such as images of a small bird in a large blue sky, dominate the reported memorizations.

To counteract this issue, Carlini et al. (2023) rescale the $\ell_2$ distance of a generation based on its relative distance from the set $\mathbb{S}_{\hat{\boldsymbol{x}}_0}$ of $\hat{\boldsymbol{x}}_0$'s $n$ nearest neighbors in $\boldsymbol{X}$. Namely, for $\mathbb{S}_{\hat{\boldsymbol{x}}_0} \subseteq \boldsymbol{X}$ and $|\mathbb{S}_{\hat{\boldsymbol{x}}_0}| = n$, we have that

$$\forall_{\boldsymbol{x} \in \boldsymbol{X} \setminus \mathbb{S}_{\hat{\boldsymbol{x}}_0}} \ell_2(\hat{\boldsymbol{x}}_0, \boldsymbol{x}) \geq \max_{y \in \mathbb{S}_{\hat{\boldsymbol{x}}_0}} \ell_2(\hat{\boldsymbol{x}}_0, \boldsymbol{y}) \,.$$

They then define the modified $\ell_2$ distance as

$$\bar{\ell}_2(\hat{\boldsymbol{x}}_0, \boldsymbol{X}; \mathbb{S}_{\hat{\boldsymbol{x}}_0}) = \frac{\ell_2(\hat{\boldsymbol{x}}_0, \boldsymbol{x})}{\alpha \cdot \mathbb{E}_{\boldsymbol{y} \in \mathbb{S}_{\hat{\boldsymbol{x}}_0}}[\ell_2(\hat{\boldsymbol{x}}_0, \boldsymbol{y})]} \,,$$

where $\alpha$ is a scaling factor. This distance decreases when $\hat{\boldsymbol{x}}_0$ is much closer to its nearest neighbor when compared to its $n$ nearest neighbors, potentially indicative of memorization.

Following their setting, we conducted experiments using DDPMs pretrained on CIFAR-10 (Krizhevsky, 2009). See Appendix Section A for implementation details. In Figure 2, we show examples of the strongest memorizations reported by $\bar{\ell}_2$ distance, demonstrating that the measure still reports monochromatic images as false positives. Most of the reported memorizations were only classified as such because they are blurry and monochromatic (which gives them an easier time matching other monochromatic images in the training set). Crucially, though, these images are *not* memorizations, because they do not contain any specifically recreated image features unique to the training set (Naseh et al., 2023). Because of this lack of specificity, we found that their metric was not a satisfying solution to detect pixel-level memorization. We apply more scrutiny to memorization metrics based on $\ell_2$ distance, as this bias towards monochromatic images has proven remarkably difficult to thoroughly eliminate.

**Embedding-Based Similarity.** Although pixel-wise distances present an intuitive approach for detecting pixel-level memorization, they are not as tailored towards reconstructive memorization. Instead, for the reconstructive case where semantic similarity is more relevant, perceptual similarity

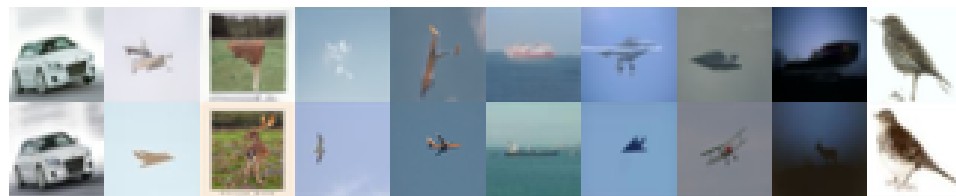

Figure 2: $\bar{\ell}_2$ **distance reports monochromatic images as memorizations.** Despite not being memorizations of their nearest neighbors in the training set, monochromatic images generate a low $\bar{\ell}_2$ distance. (**Top**) Out of 5,000 generations, the 10 generations with smallest patched $\bar{\ell}_2$ distance from CIFAR-10 train. (**Bottom**) The corresponding nearest neighbors in CIFAR-10 train to the top row of generations.

measures based on models such as SSCD (Pizzi et al., 2022), DINO (Caron et al., 2021), and CLIP (Radford et al., 2021) are often used (Somepalli et al., 2023a; Carlini et al., 2023). These metrics are generally structured with dot product similarities in a semantic embedding space, such as:

$$\sigma(\hat{\boldsymbol{x}}_0, \boldsymbol{x}) = \langle E(\hat{\boldsymbol{x}}_0), E(\boldsymbol{x}) \rangle$$

where $E(\boldsymbol{x})$ represents the embedding of an image $\boldsymbol{x}$ generated by a deep visual encoder. Perceptual metrics are robust to slight perturbations of training images such as small perspective changes. Although they perform well with reconstructive memorization, models like DINO suffer with detecting pixel-level memorization (Somepalli et al., 2023a).

One important quality of a memorization metric is the ability to remain effective and precise across different datasets. Unfortunately, past works (Carlini et al., 2023) have seen issues when attempting to translate perceptual metrics that work on Stable Diffusion to other datasets. Therefore, although the literature denotes SSCD as the standard metric for detecting reconstructive memorization (Somepalli et al., 2023a; Chen et al., 2024), it should likely only be used with datasets such as LAION-5B (Schuhmann et al., 2022) or ImageNet (Deng et al., 2009) that fit its training dataset.

### 3.2 INSPECTING SCORING STRATEGIES

Until now, we have only discussed the importance of using a consistent and reliable distance measure. It is just as important to use a scoring function that is sensitive to overall changes in memorization and does not fluctuate with unrelated changes in the model. Three strategies to aggregate a set of distances into a score include:

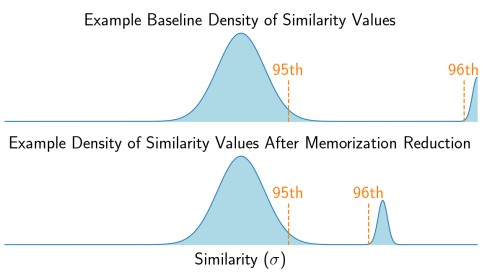

Figure 3: **95th percentile scoring fails to capture fine-grained reductions in memorization.** The above graphs demonstrate how a 95th percentile metric can fail to report successful memorization reduction. (**Top**) A distribution showing the density (vertical axis) of different similarity values (horizontal axis) in a model's baseline results. (**Bottom**) The memorization-reduced evaluation, where the 95th percentile did not change at all despite clear memorization reductions shown in the 96th percentile.

(i) the 95th percentile of similarities, (ii) the maximum similarity value, and (iii) eidetic metrics. Recently, 95th percentile scoring was employed in Somepalli et al. (2023b), where the 95th percentile of SSCD similarities was used as a metric for a number of memorization mitigation techniques. Subsequent work (Chen et al., 2024), however, questioned the validity of percentile-based scoring strategies in memorization metrics, especially when the returned distribution of distances is heavy-tailed. Figure 3 shows an example where a percentile metric could misrepresent a distribution of similarities. As a remedy, Chen et al. (2024) propose two alternatives. They recommend tracking (i) the maximum of all similarities and (ii) the number of similarities that lie above a certain threshold, a scoring idea introduced in Carlini et al. (2023). Using the maximum of all similarities could be susceptible to outliers and may not necessarily be representative of large scale trends in the similarity distribution. On the other hand, recording the number of similarities above a threshold $\delta$, also known as *eidetic memorization*, has proved to be effective. Importantly, existing literature (Somepalli et al.,

2023b; Chen et al., 2024; Kumari et al., 2023b) uses eidetic metrics with only one threshold. The problem with single threshold methods is that they do not probe how the distribution of similarities could be concentrated. Instead, multiple values for $\delta$ should be tracked to avoid flawed analysis. We elaborate on this point below in our experiments.

## 4 SOLIDMARK: A METHOD TO EVALUATE PER-IMAGE MEMORIZATION

**Motivation.** Performance on inpainting tasks significantly increases for memorized images (Carlini et al., 2023; Daras et al., 2024). Therefore, we choose inpainting as the foundation of our method. This task also stands out because of its ability to naturally function as a key-query mechanism: by masking out part of a training image, we can provide the unmasked portion to the model as a 'query' and ask it to recall the 'key' (the masked portion) from memory. Yet, two issues need consideration:

First, with inpainting, the key is almost definitely semantically related to the query, meaning the model still has a good chance to infer the masked portion of an unseen image. Additionally, the amount of useful information in the unmasked portions of different images may vary significantly, making it difficult to develop a general baseline for the model's performance on an unmemorized image. That is, it would be harder to inpaint a memorized complex image than certain unmemorized simple images. Second, since the key for inpainting is essentially a smaller image, the problems with earlier distance metrics could just propagate. For example, relatively accurate inpaintings might still produce high $\ell_2$ distances for various reasons.

**Method Structure.** To solve these issues, we assign a random scalar key to each image and embed it as the intensity of a grayscale border around that training image. By training the model on these augmented images, we teach it to output the correct grayscale intensity in the borders of an image, if memorized (see Fig. 1). Since the keys and queries are unrelated, the model outputs random grayscale borders from the distribution of the training keys for an unmemorized image.

At evaluation time, we prompt the diffusion model to outpaint the border (key) for a training image using the training caption as the text prompt and evaluate its accuracy with a scalar distance function between the grayscale intensities. This strategy solves both of our previous issues: First, since the key is unrelated to the query, we minimize the probability of inferring the key by chance. Second, since our key is a scalar, we can directly use a scalar distance function between keys (grayscale border intensities) instead of using a pixel distance function. We refer to this distance as $\ell_{\text{SM}}$ (SM = SOLIDMARK). We provide pseudocode and explain all of SOLIDMARK's hyperparameters in Appendix Section C.

## 5 EVALUATION

**Initial Validations** Since visual transformer models have been shown to pay extra attention to the center of images (Raghu et al., 2021), we were concerned that keeping the patterns as borders would uncover less memorizations than a centered pattern. For this reason, we ablated for the position of the pattern on STL-10 (Coates et al., 2011), for which the results are in Table 2. Although centered patterns did perform slightly better, we still choose to use border patterns, since the performance benefit is not worth the intrusiveness to the image generation. Implementation details for the border-center ablation are in Appendix Section D. We also validate in Appendix Section E that SOLIDMARK is able to evaluate memorization in unconditional models.

### 5.1 THE ROLE OF DATA DUPLICATION

One important concern about SOLIDMARK is that, since its keys are completely random, it sees no association between duplicated images in the training set (duplicate images will have different border colors). For this reason, one may worry that it could fail to capture memorization induced by data duplication, which is one of the most important contributing factors to memorization (Somepalli et al., 2023b). Following this concern, we introduced a large amount of exact data duplication into LAION-5K, a randomly sampled 5,000 image subset of LAION-400M (Schuhmann et al., 2021). Next, we assigned each of these duplicated images independent random keys to mimic how they

Table 2: **Reported Memorizations by SOLIDMARK and Center Patterns.** Solid patterns in the center of the image (denoted here as $\ell_{CM}$) are slightly more thorough in detecting memorization than solid borders, which is evidenced by the higher number of reported memorizations out of 10,000 images compared to $\ell_{SM}$. Lower memorization numbers indicate better model behavior. Implementation details are in Appenddix Section D.

| Metric | $\delta = 0.1$ | $\delta = 0.05$ | $\delta = 0.005$ |
|---|---|---|---|
| $(\ell_{CM}, \delta)$-Eidetic Memorizations | 1927 | 1011 | 107 |
| $(\ell_{SM}, \delta)$-Eidetic Memorizations | 1879 | 977 | 81 |

Table 3: **Reported Memorizations with Increasing Duplication.** SOLIDMARK is able to detect increased memorization in models as a response to increased duplication in the training set, even if the duplicates are assigned different keys. This is evidenced by an increase in the percentage of images reported as memorized at all eidetic thresholds $\delta$ as we increase the number of instances of duplicated images in the training set. Higher percentages indicate more memorizations. Implementation details are in Appendix Section F.

| Replications of Training Example | $\delta = 0.1$ | $\delta = 0.05$ | $\delta = 0.005$ |
|---|---|---|---|
| 2 Instances | 50% | 36% | 10% |
| 3 Instances | 56% | 60% | 26% |
| 4 Instances | 56% | 56% | 24% |
| 5 Instances | 68% | 72% | 36% |

would receive different keys in practice. We then finetuned SD 2.1 on this subset and evaluated the percentage of images for which SOLIDMARK reported memorization of at least one of its respective keys. Table 3 shows that SOLIDMARK still reports increased memorization as training set duplication increases. Implementation details are in Appendix Section F.

## 5.2 HOW FINE-GRAINED IS SOLIDMARK?

In order to understand the cues the model uses to construct memorized borders, we evaluate whether the information the model utilizes is based on the semantics of the image or on more fine-grained pixel-exactness. We evaluated changes in reported memorization as a response to small perturbations applied to the query image. To do this, we augmented LAION-5K with SOLIDMARK's borders and finetuned SD 2.1 on the augmented dataset. At evaluation time, we applied different augmentations like cropping, rotation, or blurring to the query image and observed changes in the model's memorization performance. Examples of these augmentations are in Figure 5.2 with implementation details in Appendix Section G. Overall, our results in Table 5 show that even minor perturbations to query images significantly disrupt the model's ability to recall the border color, especially when the required accuracy $\delta$ is small. These changes are not semantically meaningful and are sometimes barely visually perceptible. For this reason, we classify SOLIDMARK's reported memorizations as instances of fine-grained pixel-level memorization.

## 5.3 RE-EVALUATING MITIGATION TECHNIQUES

We use SOLIDMARK to evaluate the degree of pixel-level memorization mitigation, achievable with inference-time memorization reduction techniques. We sourced these mitigation techniques, which are described in Appendix Section H, from Somepalli et al. (2023b). For our evaluation, we augmented LAION-5K with our solid borders and finetuned SD 2.1 on this augmented dataset. We then compared the percentage decrease in $(\ell_{SM}, 0.01)$-eidetic memorizations in our model against the percentage decrease of 95th percentile SSCD similarities observed in Somepalli et al. (2023b). See Table 4 for these results. Overall, we did not find that any of the mitigation techniques that we tried significantly reduced memorization as measured by SOLIDMARK. These results are corroborated by our results in Appendix Section I. We propose this difference exists because SOLIDMARK is an

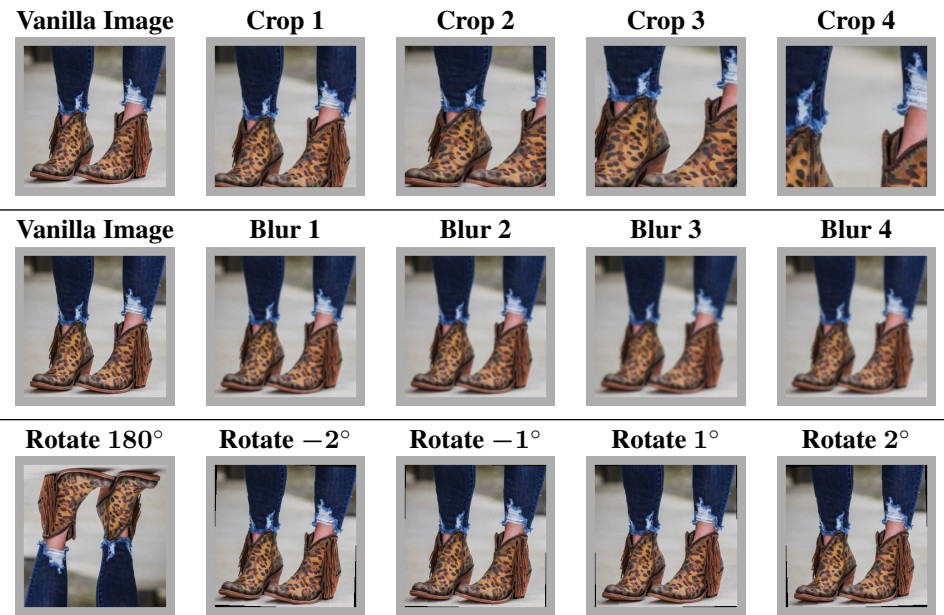

Figure 4: **Augmentations Applied to Query Images.** We show examples of the augmentations used to validate SOLIDMARK's fine-grainedness. Implementation details in Appendix Section G.

Table 4: **Evaluating Inference-Time Mitigation Methods with SOLIDMARK.** We re-evaluate several inference-time memorization mitigation methods from Somepalli et al. (2023b). SOLID-MARK reports no meaningful difference from baseline in the number of observed memorizations, which starkly contrast from the reductions reported by SSCD Similarity in the source paper. These results indicate that reconstructive memorization likely arises more from cues in the prompt compared to pixel-level memorization. Higher reduction percentages indicate that the mitigation techniques are performing better. We provide descriptions of these methods in Appendix Section H.

| Metric | GNI | RT | CWR | RNA |
|---|---|---|---|---|
| 95th Percentile of SSCD Similarities | 3.62% ↓ | **16.29%** ↓ | 9.20% ↓ | 14.33% ↓ |
| $(\ell_{SM}, 0.1)$-Eidetic Memorizations | 0.00% | 0.56%↑ | 0.75%↑ | **2.81%**↓ |
| $(\ell_{SM}, 0.05)$-Eidetic Memorizations | 5.83%↓ | 1.82%↓ | 2.91%↓ | **6.56%**↓ |
| $(\ell_{SM}, 0.005)$-Eidetic Memorizations | **3.64%**↓ | 5.45%↑ | 1.82%↑ | 0.00% |

evaluation method primarily led by visual cues in its query image. Perturbations to the queries that could, at best, dilute or change the semantic meaning of the prompt, lack a profound effect on the model's performance when the dominant visual cues are still present.

## 5.4 PRETRAINING A FOUNDATION MODEL

To foster the usage of SOLIDMARK, we pretrain and release a foundation model injected with our borders. We trained a fresh initialization of SD 2.1 on a 200k subset of LAION-5B for 500k steps. Since the model was trained with a batch size of 8, it saw every sample in this subset every 25,000 steps. Altogether, the model saw each sample 20 times. Some sample generations from this model are in Figure 5.4. Further implementation details are in Appendix Section J.

## 6 DISCUSSION

**Evaluating Individual Images.** SOLIDMARK is unique among memorization metrics in its ability to directly evaluate specific training images. In a traditional setting, one would need to repeatedly

Table 5: **Memorizations Reported with Increasing Augmentation Strength.** We show that SOLIDMARK, especially as $\delta$ decreases, reports extremely fine-grained memorizations. As we apply random cropping, rotation, and blurring to query images, the model's key prediction accuracy, measured by the number of reported $(\ell_{SM}, \delta)$-eidetic memorizations, significantly deteriorates. Higher reduction percentages indicate that the model is struggling to recognize the augmented images.

| Random Crop Strength | $\delta = 0.1$ | $\delta = 0.05$ | $\delta = 0.005$ |
|---|---|---|---|
| Baseline (0) | 1085 | 557 | 68 |
| 1 | 1038 (4.33% ↓) | 549 (1.44% ↓) | 58 (14.71% ↓) |
| 2 | 1060 (2.30% ↓) | 541 (2.87% ↓) | 48 (29.41% ↓) |
| 3 | 1035 (4.61% ↓) | 552 (0.90% ↓) | 49 (27.94% ↓) |
| 4 | 1042 (3.96% ↓) | 528 (5.21% ↓) | 50 (26.47% ↓) |
| **Rotation Angle** | | | |
| Baseline (0°) | 1085 | 557 | 68 |
| −2° | 1029 (5.16% ↓) | 506 (9.16% ↓) | 51 (25.00% ↓) |
| −1° | 1053 (2.95% ↓) | 540 (3.05% ↓) | 59 (13.24% ↓) |
| 1° | 1008 (7.10% ↓) | 502 (9.87% ↓) | 51 (25.00% ↓) |
| 2° | 1047 (3.50% ↓) | 528 (5.21% ↓) | 45 (33.82% ↓) |
| 180° | 1044 (3.78% ↓) | 522 (6.28% ↓) | 47 (30.88% ↓) |
| **Gaussian Blur Strength** | | | |
| Baseline (0) | 1085 | 557 | 68 |
| 1 | 1049 (3.32% ↓) | 516 (7.36% ↓) | 42 (38.24% ↓) |
| 2 | 1064 (1.94% ↓) | 503 (9.69% ↓) | 45 (33.82% ↓) |
| 3 | 1089 (0.37% ↑) | 534 (4.13% ↓) | 53 (22.06% ↓) |
| 4 | 1033 (4.79% ↓) | 506 (9.16% ↓) | 62 (8.82% ↓) |

**Sunny redwood forest**    **A warm fireplace**    **A long dress**    **A pair of curtains**

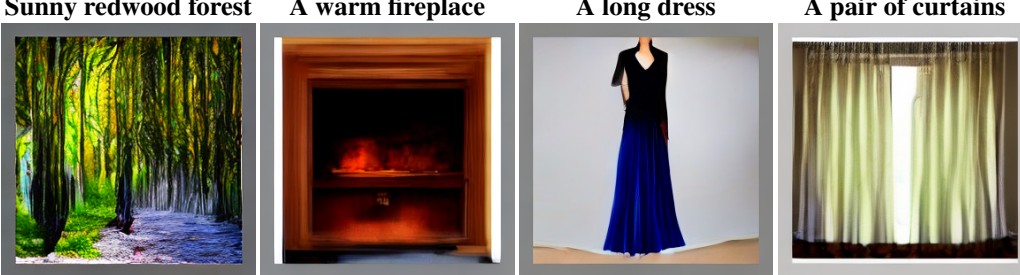

Figure 5: **Samples from Pretrained Text-to-Image Model.** (**Top**) Prompts used to generate images from our pretrained model. (**Bottom**) The resultant images for the respective prompt.

prompt a model and randomly encounter a training image to decide that it was memorized. This is problematic because prompting the model repeatedly with a very common training caption has a low chance of reproducing a given target image. Additionally, in unconditional models, which have been shown to memorize sensitive medical imaging data (Dar et al., 2024), there is no direct way to guide the output towards a specific image. SOLIDMARK, in both cases, provides an effective method to test for the memorization of specific images. In addition, it provides a continuous measure of "how memorized" an image is.

**Limitations.** By the nature of the difficulty of the setting, our method may not report memorizations that are not strong enough to capture the key. Additionally, our evaluation method has a false positive probability based on the chance of an unmemorized color randomly fitting to the key of a specific image. Additionally, SOLIDMARK may struggle with accurately reporting memorization caused by excessive exact duplication. For this reason, we encourage its use in tandem with other metrics. For an in-depth guide on how we recommend choosing a metric for a specific use case, see Appendix Section K.

## Reproducibility Statement

We include our source code for this project in the supplementary materials and release a variety of trained models to ensure that reviewers and readers can try out SOLIDMARK for themselves.

## Impact Statement

We introduce SOLIDMARK as a non-intrusive framework that can help developers evaluate and study memorization in their models. With our recommendations for how memorization metrics should be built, we hope to foster discussion about how existing metrics can be improved upon, interpreted, and generalized. Altogether, more robust evaluation of generative models helps mitigate negative privacy outcomes owing to uncaught memorization.

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

## A $\bar{\ell}_2$ DISTANCE IMPLEMENTATION DETAILS

For our experiments, we trained class-conditional DDPMs for 500 epochs on CIFAR-10 train and sampled 5,000 images with random classes, recording each generation's $\bar{\ell}_2$ distance from the training set with $n = 50$ and $\alpha = 0.5$ as in the original paper. We found that the generations were primarily monochromatic, as is shown in Figure 6.

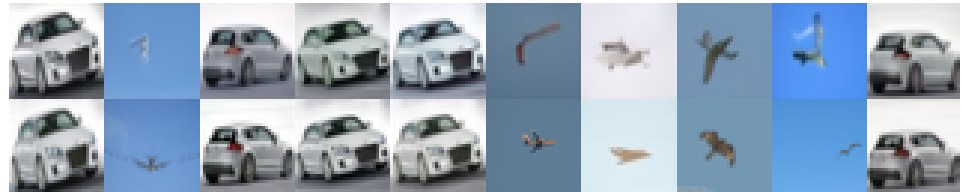

Figure 6: **Unpatched $\bar{\ell}_2$ distance reports monochromatic images as memorizations.** (**Top**) Out of 5,000 generations, the 10 generations with smallest patched $\bar{\ell}_2$ distance from CIFAR-10 train. (**Bottom**) The corresponding nearest neighbors in CIFAR-10 train to the top row of generations.

To further remedy this, we patched up the generations, which was shown to remedy this issue in (Carlini et al., 2023). We split each training image and generation into $4 \times 4$ patches. We again took $\bar{\ell}_2$ distance as our metric, except that the distance between a training image and a generation was the maximum $\ell_2$ distance between any pair of patches, one from the training image and one from the generation. The strongest memorizations with this patched distance were reported in Figure 2.

## B LDM-SPECIFIC REPAINT

We adapt the inpainting algorithm from (Lugmayr et al., 2022), which originally involves masking in a noised version of the known portion of an image at every timestep $t$ of the diffusion process. The diffusion process is thus run as normal for the unknown portion of the image. However, this kind of remasking would be impractical to do in the latent space in which LDMs operate. Instead, we chose to decode the latent (with decoder $\mathcal{D}$) every $s = 10$ steps, apply this remasking, and re-encode the latent (with encoder $\mathcal{E}$). This allows us to achieve a similar effect to RePaint without too much computational overhead. Algorithm 1 describes our adapted outpainting algorithm for a model $\epsilon_\theta$ to outpaint an image $\boldsymbol{x}$ with conditional (prompt) embedding $\boldsymbol{c}$ and mask $m$.

## C SOLIDMARK IMPLEMENTATION DETAILS

Given an image-text pairs dataset, we augment each data point with these borders. This new dataset can be used to either train from scratch or finetune a diffusion model. To evaluate memorization for

---

**Algorithm 1** Adapted Outpainting Algorithm for LDMs

---

**procedure** OUTPAINT($\epsilon_\theta, \boldsymbol{x}, m$)
 $\boldsymbol{z}_T \sim \mathcal{N}(\mathbf{0}, \mathbf{I})$
 **for** $t = T$ to $1$ **do**
  $\tilde{z} \sim \mathcal{N}(\mathbf{0}, \mathbf{I})$ **if** $t > 1$ **else** $\tilde{z} \leftarrow 0$
  $\boldsymbol{z}_{t-1} \leftarrow \frac{1}{\sqrt{\alpha_t}}\left(\boldsymbol{z}_t - \frac{\beta_t}{\sqrt{1-\bar{\alpha}_t}}\epsilon_\theta(\boldsymbol{z}_t, \boldsymbol{c}, t)\right) + \sigma_t\tilde{z}$  $\triangleright \sigma_t, \beta_t, \alpha_t, \bar{\alpha}_t$: Noise/Variance schedule
  **if** $t \bmod s = 0$ **then**           $\triangleright s$: Precision hyperparameter
   $\epsilon \sim \mathcal{N}(\mathbf{0}, \mathbf{I})$
   $\boldsymbol{x}_{t-1}^{\text{known}} \leftarrow \sqrt{\bar{\alpha}_t}\boldsymbol{x}_0 + (1 - \bar{\alpha}_t)\epsilon$
   $\boldsymbol{x}_{t-1}^{\text{unknown}} \leftarrow \mathcal{D}(\boldsymbol{z}_{t-1})$         $\triangleright \mathcal{D}$: Visual Decoder
   $\boldsymbol{x}_{t-1} \leftarrow m \odot \boldsymbol{x}_{t-1}^{\text{unknown}} + (1 - m) \odot \boldsymbol{x}_{t-1}^{\text{known}}$
   $\boldsymbol{z}_{t-1} \leftarrow \mathcal{E}(\boldsymbol{x}_{t-1})$          $\triangleright \mathcal{E}$: Visual Decoder
  **end if**
 **end for**
 **return** $\mathcal{D}(\boldsymbol{z}_0)$
**end procedure**

---

a single training image $\boldsymbol{x}$ with conditional embedding $\boldsymbol{c}$, we prompt the model to outpaint $\boldsymbol{x}$'s border as in Lugmayr et al. (2022)[2], using $\boldsymbol{x}$'s corresponding caption to condition the model. We evaluate a generation $\hat{\boldsymbol{x}}$ against the true key $k_{\boldsymbol{x}}$ for $\boldsymbol{x}$ by finding the absolute difference between the true key $k_{\boldsymbol{x}}$ and the average of $\hat{\boldsymbol{x}}$'s border pixels (the model's "predicted key"). For a full model evaluation, we calculate the eidetic memorizations for every image in a subset of the training data, where the accuracy of the evaluation increases with the size of the subset.

Formal pseudocode to use SOLIDMARK to evaluate a model $\epsilon_\theta$ is in Algorithm 2. Our method has a few hyperparameters to consider: the keymap $k(\boldsymbol{x})$, pattern thickness $p$, the number $n$ of training samples that are evaluated, and the number of times $r$ each sample was evaluated (usually set to 1 unless $n$ is small). We constructed our keymaps by assigning random floats to each image: $k_{\boldsymbol{x}} \sim$ Unif$(0, 1)$; we draw a grayscale color (which is the same scalar across all channels) uniformly at random (assuming images are representing with floating points from 0-1). We always used a pattern thickness of 16. When evaluating on a model scale, we evaluated on $n = 10,000$ or $n = 5,000$ samples in all cases. When we did not have so many samples, we compensated by increasing $r$ to stabilize the results.

## D    BORDER-CENTER ABLATION IMPLEMENTATION DETAILS

We compared the memorization evaluation performance of borders of thickness 16 against $16 \times 16$ center patterns. To do this, we combined STL-10's labelled train and validation sets to form a training set of $13,000$ images. We augmented the training set with the respective patterns and pretrained class-conditioned DDPMs on these training sets for 500 epochs with 250 sampling steps and a batch size of 64. Afterwards, we evaluated random $10,000$ image subsets from both models and reported the results in Table 2.

## E    UNCONDITIONAL MODELS

We pretrained an unconditional DDPM on CelebA (Liu et al., 2015) augmented with SOLIDMARK's borders for 225 epochs with a batch size of 64 and 250 sampling steps. Afterwards, we used SOLIDMARK to evaluate memorization within the model over 5,000 images. The results of this evaluation are in Table 6.

---

[2]We describe a modified version of this algorithm that we use for LDMs in Appendix Section B.

---

**Algorithm 2** Using SOLIDMARK for Evaluation

---

**procedure** TRAINWITHSOLIDMARK($\epsilon_\theta, \boldsymbol{X}, k, p$)
    $\bar{\boldsymbol{X}} \leftarrow [\,]$
    **for** $\boldsymbol{x}, \boldsymbol{c} \in \boldsymbol{X}$ **do**                     ▷ Iterate over images and captions in the dataset.
        $k_{\boldsymbol{x}} \leftarrow k(\boldsymbol{x})$
        Append grayscale border with magnitude $k_{\boldsymbol{x}}$ and thickness $p$ to $\boldsymbol{x}$
        $\bar{\boldsymbol{X}} \leftarrow \bar{\boldsymbol{X}} + [\boldsymbol{x}]$
    **end for**
    TRAINMODEL($\epsilon_\theta, \bar{\boldsymbol{X}}$)
**end procedure**

**procedure** ISIMAGEMEMORIZED($\epsilon_\theta, \boldsymbol{x}, \boldsymbol{c}, \boldsymbol{\delta}, k, p, r$)
    $k_{\boldsymbol{x}} \leftarrow k(\boldsymbol{x})$                           ▷ Same $k_{\boldsymbol{x}}$ assigned to $\boldsymbol{x}$ during training
    $m \leftarrow$ a mask of size equal to augmented $\boldsymbol{x}$: 0 everywhere except for a $p$-thick border of 1.
    **for** $i = 1$ to $r$ **do**
        $\hat{\boldsymbol{x}}_0 \leftarrow$ OUTPAINT($\epsilon_\theta, \boldsymbol{x}, \boldsymbol{c}, m$)             ▷ Outpaint on $\boldsymbol{x}$ to yield generation $\hat{\boldsymbol{x}}_0$.
        $\hat{k} \leftarrow \sum(m \odot \hat{\boldsymbol{x}}_0) / \sum(m)$         ▷ Predicted key $\hat{k}$ is the average of $\hat{\boldsymbol{x}}_0$'s border.
        **if** $|\hat{k} - k| \leq \boldsymbol{\delta}$ **then**
            **return** True                  ▷ Image is $(\ell_{\text{SM}}, \boldsymbol{\delta})$-eidetically memorized.
        **end if**
    **end for**
    **return** False                          ▷ Image was not recognized.
**end procedure**

**procedure** EVALUATEMODEL($\epsilon_\theta, \bar{\boldsymbol{X}}, \boldsymbol{\delta}, n, k, p, r$)
    $\bar{\boldsymbol{X}}_n \leftarrow$ Size $n$ random subset of $\bar{\boldsymbol{X}}$
    Mems $\leftarrow 0$
    **for** $\boldsymbol{x}, \boldsymbol{c} \in \bar{\boldsymbol{X}}_n$ **do**
        **if** ISIMAGEMEMORIZED($\epsilon_\theta, \boldsymbol{x}, \boldsymbol{c}, \boldsymbol{\delta}, k, p, r$) **then**
            Mems $\leftarrow$ Mems+1
        **end if**
    **end for**
    **return** Mems
**end procedure**

---

Table 6: **Reported Memorizations in Pretrained CelebA Model.** We show the number of $(\ell_{\text{SM}}, \boldsymbol{\delta})$-eidetic memorizations found in a 5,000 image subset of CelebA's train set over a few values for $\boldsymbol{\delta}$. This demonstrates that SOLIDMARK is able to measure memorization in unconditional models.

| $\boldsymbol{\delta} = 0.1$ | $\boldsymbol{\delta} = 0.05$ | $\boldsymbol{\delta} = 0.005$ |
|:---:|:---:|:---:|
| 924 | 466 | 37 |

## F  DATA DUPLICATION IMPLEMENTATION DETAILS

We duplicated 50 images from LAION-5K 2, 3, and 4 times respectively and finetuned SD 2.1 on this dataset for 50 epochs. At evaluation time, we evaluated each image 10 times and reported whenever an image was classified as a $(\ell_{\text{SM}}, \boldsymbol{\delta})$-eidetic memorization of any of its respective keys in the training dataset.

## G  AUGMENTATION ABLATION IMPLEMENTATION DETAILS

For increasing crop levels, we altered the scale at which random cropping operated. For crop level 1, we randomly cropped the image to relative size (0.8, 0.8) and resized it to its original size. For crop level 2, we cropped to size (0.6, 0.6). We used (0.4, 0.4) for crop level 3 and (0.2, 0.2) for crop

level 4. For different blur levels, we used Gaussian blurring with a kernel size of (5, 5) for blur level 1, (9, 9) for blur level 2, (13, 13) for blur level 3, and (17, 17) for blur level 4.

## H  EVALUATED INFERENCE-TIME MITIGATION METHODS

We evaluated Gaussian Noise at Inference (GNI), Random Token Replacement and Addition (RT), Caption Word Repetition (CWR), and Random Numbers Addition (RNA). GNI adds a small amount of random noise to text embeddings, usually from a distribution of $\mathcal{N}(0, 0.1)$. To tune this method, we increased the magnitude of the perturbations in order to reduce memorization further at the cost of adherence to the conditional prompt. RT randomly replaces tokens in the caption with random words and adds random words to the caption. On each iteration, RT has a chance to randomly replace each individual token in the prompt; this method was tuned by changing the number of iterations. CWR randomly chooses a word from the given prompt and inserts it into one additional random spot in the prompt. RNA, at each iteration, randomly adds random numbers in the range $\{0, 10^6\}$ to the prompt, hoping to perturb the prompt without changing its semantic meaning. Similar to RT, CWR and RNA were tuned by changing the number of iterations.

## I  GNI EVALUATION IN DDPMS

We augmented STL-10 (Coates et al., 2011) with a 16-thick border and pretrained DDPMs on this augmented dataset. Next, we added Gaussian noise to the conditional embeddings with mean 0 and a range of magnitudes, tracking the number of $(\ell_{\text{SM}}, \boldsymbol{\delta})$-eidetic memorizations over $5,000$ generations as the magnitude of noise increased. These results are in Figure 7. Overall, we found that for both values of $\boldsymbol{\delta}$, the number of $(\ell_{\text{SM}}, \boldsymbol{\delta})$-eidetic memorizations remained relatively constant.

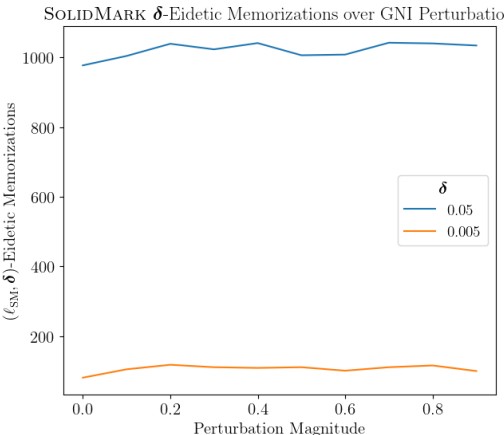

Figure 7: **Reported Memorizations in DDPM Pretrained on STL-10 with GNI.** We report the change in the number of memorizations while applying progressively increasing magnitudes of GNI to the model. Overall, we find no significant change in the number of memorizations at any magnitude of noise, supporting our argument that visual cues are more closely tied to pixel-level memorization than conditional embeddings.

## J  PRETRAINED TEXT-TO-IMAGE MODEL IMPLEMENTATION DETAILS

We used a random 200k subset of LAION-5B to train our model. This model trained for 500k steps with a batch size of 8 over 4 days and 10 hours on 4 H100 GPUs. We only pretrained the UNet, which was a fresh initialization of SD 2.1's UNet. We trained this model with the HuggingFace StableDiffusionPipeline. All samples were taken using 250 sampling steps.

## K  How to Choose a Metric

Given the outlined qualities of a stable metric, we suggest that eidetic scoring always be used, regardless of the choice of distance function. A few values of $\delta$ should be chosen and tracked simultaneously, especially if a memorization reduction technique is being applied (to give an idea of how fine-grained the changes in memorization are).

If reconstructive memorization is to be tracked, one should first consider the training dataset. If a large dataset of natural images such as LAION-5B (Schuhmann et al., 2022) was used to train the model, then, following the literature, SSCD similarity is likely the most consistent and precise metric. If a smaller or more niche dataset is being used, then the optimal choice of distance function is a comparatively unexplored question. Intuitively, two suggestions could be to pretrain a self-supervised model on the dataset or to finetune SSCD on the dataset. Whichever approach is chosen, it is important to evaluate samples by hand and ensure that the metric is sufficiently specific and sensitive.

For pixel-level memorization, one might feel inclined to only use an $\ell_2$-based metric because of their simplicity. This approach is problematic, though, as $\ell_2$-based metrics are not robust against small shifts in pixel space and tend to report false positives with their biases towards monochromatic images. We instead recommend that SolidMark be used in this situation. If the dataset in question contains lots of exact duplication, then an $\ell_2$-based metric should be used in tandem with SolidMark while manually validating the $\ell_2$ metric's reported memorizations. This way, no memorizations will be missed, but the evaluation will remain fine-grained.

