# OpenReview forum: "SolidMark: Evaluating Image Memorization in Generative Models"
_ICLR.cc/2025/Conference — ICLR 2025 Conference Withdrawn Submission_

### Official Review · Reviewer_VbaQ · 2024-10-31

**Soundness:** 2
**Presentation:** 3
**Contribution:** 2
**Rating:** 5
**Confidence:** 4

**Summary:**

This paper proposes a kind of evaluation method that measures the memorization of a generative model. The method injects random keys into training images and tests how many training keys can be reproduced at inference.

**Strengths:**

+ The paper is well-motivated, with good logic. The analyses of the existing methods are convincing.
+ The proposed *outpainting-like* method is novel, interesting, and easy to perform.

**Weaknesses:**

- The paper uses 5,000 images as the training set (am I correct?) . I think the training set size is too small, and is easily memorized with sufficient long learning by large models such as SD 2 . What I am concerned about is what proportion of data is memorized when training with a huge set.
- This method seems to only work for generative models that can be fine-tuned as an in/outpainting model.
- Since the model has been fine-tuned, is it still capable of reflecting the memorization of the original model?
- Although the approach is novel and interesting, it lacks strong evidence to support its effectiveness as an evaluation method for memorization.

**Questions:**

See the "Weaknesses".

---

### Official Review · Reviewer_pUDF · 2024-10-31

**Soundness:** 1
**Presentation:** 2
**Contribution:** 1
**Rating:** 1
**Confidence:** 3

**Summary:**

The authors propose a new evaluation method for image memorization in image models. The core idea of their method is to augment the training images (denoted with query) with grayscale boarders at different intensities (denoted as keys). After training, the authors perform out-painting on the training images (queries) and evaluate the reconstruction of the boarders (keys). Since the query and keys are not related, the authors claim that the reconstruction of the keys indicates memorization.

**Strengths:**

- The method proposed by the authors is interesting.
- The authors discussed thoroughly the related work and previous evaluation details.
- The authors provided code and implementation details.

**Weaknesses:**

1- Most of the paper is focused on related work, with only a brief section on the proposed metric. A mathematical formulation and discussion of limitations would improve understanding.

2- The metric requires either fine-tuning or training with augmented images. The first alters the model’s behavior (for example a model trained on a large-scale dataset would not necessarily have the same memorization level as the one fine-tuned on a smaller dataset). The latter could potentially hurt performance of the generative model. Additionally, both add unnecessary computational demands.

3- Fine-tuning raises questions about dataset size and training time, adding more complexity. For example, a single image would probably lead to overfitting and model memorization, while fine-tuning on a large dataset would be impractical, making it unclear where to draw the line between the two.

4- The reconstruction capability of the proposed augmentation with grayscale borders may be misleading. Since the border is unrelated to the training image and cannot be derived from the prompt or image content, the generative model might produce a uniform border across images or treat it differently. In other words, there is no theoretical or practical gurantee that memorizing borders correlates with memorizing the actual image content.

5- The reliance on outpainting adds a layer of complexity. It’s unclear if outpainting memorization directly relates to generated image memorization. Would different out-painting methods behave differently? What if an out-painting method significantly alter the generation that hides the model memorization?

6- In general, there is no evidence supporting the validity of the metric.

7- The metric only assesses pixel-level memorization, and not semantic-level one.

**Questions:**

Please see weaknesses above.

---

### Official Review · Reviewer_EnLk · 2024-11-04

**Soundness:** 1
**Presentation:** 1
**Contribution:** 2
**Rating:** 5
**Confidence:** 3

**Summary:**

This paper introduces a new metric for evaluating the memorization of generative models. Specifically, Solidmark modifies each image with a grayscale border, make the model perform outpainting and examining the reconstruction. The metric reports the distance between the predicted key and the true value.

**Strengths:**

- The paper addresses an important issue: the memorization of generative models. It highlights the lack of a standard metric for this purpose, which I think is a crucial issue.
- An interesting observation is made (Figure 2) regarding the false positive issues associated with L2 distance.

**Weaknesses:**

- It is difficult to assess whether the proposed method is suitable for evaluating memorization. Although some effectiveness is demonstrated in the experiments, there are concerns whether the approach is technically sound. Despite reading the methods section multiple times, I still do not understand why it serves as a sign of memorization.
- To establish the validity of the proposed metric, comparisons with existing metrics, such as SSCD or L2 distance, should be included. The paper does not clarify how the proposed metric aligns with existing memorization metrics, such as those referenced in [1].

[1] Stein, George, et al. "Exposing flaws of generative model evaluation metrics and their unfair treatment of diffusion models." *Advances in Neural Information Processing Systems* 36 (2024).

**Questions:**

- Does the border have to be gray? Could other watermarking methods be used? What about augmentations other than the border? In other words, what was the rationale behind specifically choosing a gray border?
- Can it be proven that the proposed method accurately indicates true memorization? Is there a possibility that this is merely a correlation? To validate this paper, experiments should be conducted to determine whether the findings align (at least partially) with existing memorization metrics.

---

### Official Review · Reviewer_kF8o · 2024-11-04

**Soundness:** 3
**Presentation:** 2
**Contribution:** 3
**Rating:** 5
**Confidence:** 3

**Summary:**

The paper introduces SolidMark, a new method for evaluating pixel-level memorization in diffusion models. This method performs the evaluations via augmenting images with random grayscale borders and using outpainting to evaluate border reconstruction. The authors validate the proposed method through extensive experiments and comparisons with existing metrics, demonstrating its effectiveness and potential applications.

**Strengths:**

- The authors provide detailed implementation details and source code, facilitating the reproducibility of the proposed method.

- The authors conduct sufficient ablation studies to evaluate the proposed method.

**Weaknesses:**

- The presentation of this paper makes it challenging for readers unfamiliar with the memorization task to follow. The authors should polish the presentation of this paper.

- The reasons for choosing inpainting and outpainting instead of other image transformations for evaluating memorization should be discussed more thoroughly.

- The paper does not explore whether this method can be generalized to other diffusion models such as SDXL, Pixart, and Flux.

- Is this method still effective in multi-resolution scenarios? For instance, an object "A" is typically located in the center of images and the diffusion model can generate such images. Can the proposed method determine whether this diffusion model has memorized "A" given a test image where "A" is located in the upper-left area?

**Questions:**

Please refer to weakness.

---

### Note · Authors · 2024-11-15

**Comment:**

We thank all the reviewers for their dedicated efforts. Given the status of the reviews, we will revise the manuscript and submit it to another conference.

**Withdrawal Confirmation:**

I have read and agree with the venue's withdrawal policy on behalf of myself and my co-authors.